# A Lightweight nnU-Net Combined with Target Adaptive Loss for Organs and Tumors Segmentation

Tao Liu[1][0009−0007−2933−9197], Xukun Zhang[1][0000−0003−2869−9434], Minghao Han[1][0009−0002−0043−7539], and Lihua Zhang[1][0000−0003−0467−4347]

Fudan University, Shanghai 200082, China
{lihuazhang}@fudan.edu.cn

**Abstract.** Accurate and automated abdominal organs and tumors segmentation is of great importance in clinical practice. Due to the high time- and labor-consumption of manual annotating datasets, especially in the highly specialized medical domain, partially annotated datasets and unlabeled datasets are more common in practical applications, compared to fully labeled datasets. CNNs based methods have contributed to the development of medical images segmentation. However, previous CNN models were mostly trained on fully labeled datasets. So it is more vital to develop a method based on partially labeled datasets. In FLARE23, we design a model combining a lightweight nnU-Net and target adaptive loss (TAL) to obtain the segmentation results efficiently and make full use of partially labeled dataset. Our method achieved an average DSC score of 86.40% and 19.41% for the organs and lesions on the validation set and the average running time and area under GPU memory-time cure are 25.34s and 23018MB, respectively.

**Keywords:** abdominal organs and tumors segmentation · lightweight nnU-Net · target adaptive loss.

## 1 Introduction

A precise pixel-level understanding of abdominal anatomy image is of vital importance for computer-aided clinical practice such as disease diagnosis, surgery navigation, radiation therapy and so on. Specifically, accurate abdominal organs and lesions segmentation plays a fundamental role in supporting clinical workflows, including diagnostic interventions and treatment planning, which can be an essential step for preoperative diagnosis.

Thanks to the significant development of deep learning, many abdominal organ segmentation methods have been designed based on deep CNNs, such as nn-UNet and 3D-UNet, which achieve great performance on different abdominal organ datasets. However, most models typically require all organs of interest to be annotated. But, it is unrealistic to get a dataset with all organs annotated because of the time- and labor-consuming labeling process. Hence, it is still an important task to segment multi-organs based on a partial labeled dataset.

Currently, there exist also numerous studies dedicated to solving the problem of abdominal multi-organ and tumor segmentation. But these methods all have a common limitation, which is that the models they developed are limited to the segmentation of a certain organ and its lesions. When it comes to migrating these models to another organ segmentation task, it doesn't work. There are still no general models for universal abdominal organ and tumor segmentation at present. As a result, it remains a challenging task to segment multi-organs and all tumors with one model.

FLARE2023 is a competition which aims to promote the development of universal organ and tumor segmentation in abdominal CT scans. The competition organizer provided a training set including 4000 3D CT scans from over 30 medical centers, of which 2200 cases are partial labeled and 1800 cases don't have labels, and a validation set including 100 cases. In addition to precise segmentation of the 13 abdominal organs, the algorithm provided by the contestants also requires the recognition and segmentation of all the tumors on different organs in abdominal CT images, which is a challenging task. This is the first challenge which focuses on pan-cancer segmentation in CT scans. In addition, the competition also imposes limitations on inference speed, memory, and GPU memory. Each test sample needs to spend less than 28GB of memory within 60 seconds of prediction time to obtain inference result. And the peak GPU memory overhead should preferably be below 4GB, which further increases the difficulty of the competition.

We extensively investigated image segmentation methods based on partially annotated datasets, especially in medical domain. During the past several years, many studies have been devoted to solving the problem of abdominal multi organ segmentation in partially annotated datasets, but this problem remains a challenging one. A straightforward strategy is to train as many networks as partially labeled datasets, but suffers from several shortcomings including: (1) less training data for each single network, (2) longer inference time and longer training time.

Also, much more attention have been paid on training one model with several partially labeled datasets. Intuitively speaking, this strategy has many advantages, including but not limited to fully utilizing different datasets to improve robustness of model. The methods can be generally grouped into two categories. The first category is to design new network to handle this problem. Chen et al. [18] designed a network with a task-shared encoder and as many task-specific decoders as partially labeled datasets. But this kind of network has been proven to be memory-consuming. Zhang et al. [12] proposed a dynamic on-demand network (DoDNet) by catenating a one-hot vector of equal length to the number of organs with the features of images as task-specific prompt to generate weights for dynamic convolution filters. The second type of methods attempt to design adaptive loss functions that can be directly applied to partially labeled data. Fang et al. [3] proposed a target adaptive loss (TAL) to train a network on several partially labeled dataset by treating the organs with unknown labels as background. Additionally, Shi et al. [6] merged unlabeled organs with the back-

ground by imposing an constraint on each voxel of images and then propose a marginal and exclusive loss to train a model based on a fully labeled dataset and several partially labeled datasets. Furthermore, Liu et al. [7] studied the partial-label segmentation on the existing approaches and identified three distinct types of supervision signals, including two signals derived from ground truth and one from pseudo label and then they proposed a training framework called COSST, which combined comprehensive supervision signals and self-training with pseudo labels, which has been demonstrated consistent great performance.

After reviewing existing methods for abdominal multi-organ segmentation based on partially labeled datasets, inspired by Fang et al., we plan to follow their design in their work, treating unlabeled organs as background and using the target adaptive loss (TAL) function proposed in  [3]. Specifically, we merge the output channels of unlabeled organs and the original background channel into a new one. The reason for doing this is because there are always unlabeled organs in most images of the FLARE23 dataset, resulting in the inapplicability of common segmentation losses, such as dice loss. By utilizing the TAL loss, it can effectively handle this problem. What's more, due to the official requirements for segmentation efficiency and memory utilization in the competition, existing default CNNs or transformers are not competent for this task. We retrospected the top methods in FLARE22 and FLARE21, and we found that the lightweight nnU-Net designed by the top method in FLARE22 achieved remarkable efficiency without significantly reducing segmentation performance. Hence, we attempt to extend the lightweight nnU-Net proposed in FLARE22 with the target adaptive loss, to handle the segmentation of the partially labeled dataset in an efficient and effective manner.

All in all, our proposed method can be summarized as combining the lightweight nnU-Net with target adaptive loss function to achieve efficient and accurate segmentation. We will provide a detailed introduction to our proposed method in the following chapter.

## 2   Method

In this section, we will give a detailed description of our proposed method. As illustrated in Fig. 1, our proposed method is mainly based on a lightweight nnU-Net and a target adaptive loss, which is used to handle with the partially labeled dataset.

### 2.1   Preprocessing

It is vital to perform data preprocessing before training. In our proposed scheme, data preprocessing can be divided into five parts, which is:

(1) Statistical analysis: We conducted statistical analysis on the distribution of labels in the dataset and concluded that tumor labels are distributed across different organs and are unevenly distributed, making tumor segmentation tasks very difficult.

(2) Make sure the geometry of label file match with the geometry of image file. Some cases in the dataset doesn't meet this requirement, which will influence the subsequent operation.

(3) Cropping: Cropping out voxels with a value of zero in the image, which don't have useful information and don't affect the subsequent learning process. Instead, it can significantly reduce the image size and computational complexity.

(4) Resampling: Resampling is a crucial step to avoid the problem of inconsistent actual spatial sizes represented by individual voxels in different images. By default setting of nnU-Net, in anisotropic datasets, for dimension with particularly large spacing, take the 10% quantile of the spacing value of that dimension in the dataset as the target space size for that dimension.

(5) Normalization: The purpose of normalization is to ensure that the grayscale values of each image in the training set have the same distribution. The normalization operation in our method is the same as what nnU-Net does.

### 2.2   Proposed Method

Fig. 1 shows the framework of our proposed method. As illustrated in Fig. 1, our proposed method mainly composes of two parts, a lightweight nnU-Net and a target adaptive loss (TAL), of which, the lightweight nnU-Net is adapted from the top method in FLARE22 and the TAL is used for training with partial labels.

Specifically, the lightweight nnU-Net is modified based on the default nnU-Net to improve inference speed and reduce resource consumption, and the main focus is to change channels in the first stage into 16, and change convolution number per stage into 2. Additionally, it performs downsampling only twice during inference stage, and the input patch size is reduced, the input spacing is increased to obtain a low resolution of image. We don't apply any extra strategy to improve inference speed and reduce resource consumption, except for following what the top method [10] did to their small nnU-Net.

Furthermore, the target adaptive loss we use can be formulated as follow:

$$L_{TAL} = \sum_{c \in B} y_v^c \log \hat{y_v^c} + \mathbf{1}_{[\sum_{c \in B} y_v^c = 0]} \log(1 - \sum_{c \in B} \hat{y_v^c})$$

where $B$ denotes the organs labeled in the input batch, $\hat{y_v^c}$ is the predicted probability of voxel $v$ labeled as class $c$ and $y_v^c$ is from ground truth, which indicates whether voxel $v$ labeled as class $c$ or not.

We treat the unlabeled organs in images as background by merging the output channels of unlabeled organs and original background channel into a new one. And then the network can be trained with supervision by TAL.

We used the pseudo labels of the 1800 unlabeled images, generated by the FLARE22 winning algorithm [10].

### 2.3   Post-processing

We didn't use any post-processing in our method.

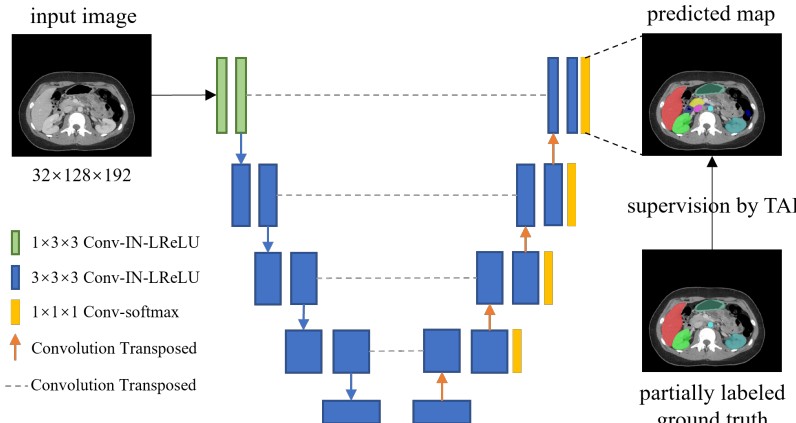

**Fig. 1.** Network architecture, which includes a lightweight nnU-Net to segment images efficiently and TAL to train model based on partially labeled dataset.

## 3   Experiments

### 3.1   Dataset and evaluation measures

The FLARE 2023 challenge is an extension of the FLARE 2021-2022 [14][15], aiming to aim to promote the development of foundation models in abdominal disease analysis. The segmentation targets cover 13 organs and various abdominal lesions. The training dataset is curated from more than 30 medical centers under the license permission, including TCIA [2], LiTS [1], MSD [19], KiTS [8,9], autoPET [5,4], TotalSegmentator [20], and AbdomenCT-1K [16]. The training set includes 4000 abdomen CT scans where 2200 CT scans with partial labels and 1800 CT scans without labels. The validation and testing sets include 100 and 400 CT scans, respectively, which cover various abdominal cancer types, such as liver cancer, kidney cancer, pancreas cancer, colon cancer, gastric cancer, and so on. The organ annotation process used ITK-SNAP [21], nnU-Net [11], and MedSAM [13].

The evaluation metrics encompass two accuracy measures—Dice Similarity Coefficient (DSC) and Normalized Surface Dice (NSD)—alongside two efficiency measures—running time and area under the GPU memory-time curve. These metrics collectively contribute to the ranking computation. Furthermore, the running time and GPU memory consumption are considered within tolerances of 15 seconds and 4 GB, respectively.

### 3.2   Implementation details

**Environment settings** The development environments and requirements are presented in Table 1.

**Table 1.** Development environments and requirements.

| | |
|---|---|
| System | Ubuntu 20.04.1 LTS |
| CPU | Intel(R) Xeon(R) CPU E5-2680 v4 @ 2.40GHz |
| RAM | 4×32GB; 2400MT/s |
| GPU (number and type) | Two NVIDIA Quadro RTX 8000 48G |
| CUDA version | 12.0 |
| Programming language | Python 3.7 |
| Deep learning framework | torch 1.12.0, torchvision 0.13.0 |
| Specific dependencies | None |
| Code | |

**Training protocols** We used the pseudo labels of the 1800 unlabeled images, generated by the FLARE22 winning algorithm [10]. As for the partial labels, We treated the unlabeled organs in images as background by merging the output channels of unlabeled organs and original background channel into a new one. Furthermore, we applied the same data augmentation, patch sampling strategy and optimal model selection criteria as the default settings of nnU-Net.

**Table 2.** Training protocols.

| | |
|---|---|
| Network initialization | |
| Batch size | 2 |
| Patch size | 32×128×192 |
| Total epochs | 1500 |
| Optimizer | SGD |
| Initial learning rate (lr) | 0.01 |
| Lr decay schedule | $(1 - epoch/1000)^{0.9}$ |
| Training time | 36 hours |
| Loss function | TAL (detailed in section 2.2) |
| Number of model parameters | 5.64M[1] |
| Number of flops | 8.13G[2] |
| $CO_2$eq | 5.3 Kg[3] |

## 4    Results and discussion

### 4.1    Quantitative results on validation set

The Dice and NSD scores of organs and tumors on the validation set is given in Table 3.

We have done ablation studies to analyze the effect of unlabelled data. We trained another same network as mentioned above, but we only used labeled

**Table 3.** Quantitative evaluation results.

| Target | Public Validation | | Online Validation | | Testing | |
|---|---|---|---|---|---|---|
| | DSC(%) | NSD(%) | DSC(%) | NSD(%) | DSC(%) | NSD (%) |
| Liver | 95.59 ± 6.67 | 90.31 ± 8.44 | 95.87 | 96.51 | 94.34 | 94.87 |
| Right Kidney | 91.25 ± 10.14 | 88.74 ± 10.13 | 90.41 | 91.93 | 92.28 | 93.55 |
| Spleen | 95.68 ± 3.71 | 94.70 ± 6.32 | 95.62 | 96.96 | 95.48 | 97.07 |
| Pancreas | 83.57 ± 7.76 | 80.23 ± 11.67 | 82.13 | 94.21 | 86.50 | 95.64 |
| Aorta | 92.78 ± 5.08 | 91.27 ± 8.07 | 94.19 | 96.96 | 90.90 | 94.44 |
| Inferior vena cava | 89.33 ± 6.68 | 83.14 ± 9.38 | 89.99 | 91.75 | 85.97 | 88.56 |
| Right adrenal gland | 82.36 ± 3.50 | 93.16 ± 3.90 | 80.97 | 94.04 | 75.62 | 88.03 |
| Left adrenal gland | 79.58 ± 9.55 | 89.94 ± 10.07 | 79.16 | 91.27 | 75.91 | 87.42 |
| Gallbladder | 83.47 ± 13.53 | 83.47 ± 13.53 | 78.99 | 78.26 | 76.84 | 78.29 |
| Esophagus | 75.82 ± 17.90 | 77.64 ± 16.59 | 79.04 | 90.48 | 83.94 | 94.11 |
| Stomach | 89.23 ± 9.50 | 83.12 ± 15.55 | 88.78 | 92.23 | 83.61 | 97.10 |
| Duodenum | 77.51 ± 10.38 | 73.11 ± 11.33 | 77.36 | 91.75 | 78.25 | 91.50 |
| Left kidney | 89.68 ± 14.61 | 87.21 ± 15.70 | 90.69 | 91.88 | 92.03 | 93.39 |
| Tumor | 23.36 ± 25.43 | 18.83 ± 21.51 | 19.41 | 12.25 | 24.88 | 14.91 |
| Average | 82.09 ± 17.42 | 81.06 ± 11.59 | 81.62 | 86.46 | 81.18 | 85.63 |

data to train this network. We divided 2200 labeled data into two equal parts, with the first 50% using official labels provided by the competition and the last 50% using pseudo labels generated by the FLARE22 winning algorithm [10]. Not surprisingly, the network model using unlabeled data performs better than the one that doesn't use. Network trained with both labeled and unlabeled data is exposed to more data during the training phase, result in stronger generalization ability. The validation results of the model trained without unlabeled data are given in Table 4.

## 4.2   Qualitative results on validation set

Fig. 2 shows four examples of segmentation results in the validation set, with two good ones and two bad ones. It can be easily seen that our method outperforms out ablation study results, which is due to the better generalization of model trained with more data. Case 0007 performed well in tumor segmentation tasks, but poorly in organ segmentation tasks. Our analysis suggests that the model may have focused more on tumors but neglected organs, and in this example, the tumor is completely located on the surface of the liver, making it difficult for the model to recognize the liver. Furthermore, we think the reason why case 0035 performed badly is that tumors spread all over left kidney, which is a hard case, causing the model to be unable to recognize left kidney and tumor. As for the two good ones, we think it may be because the location of the tumor is easier to recognize and the image is clearer.

**Table 4.** Quantitative evaluation results of the model trained without unlabeled data.

| Target | Public Validation | | Online Validation | |
|---|---|---|---|---|
| | DSC(%) | NSD(%) | DSC(%) | NSD(%) |
| Liver | 95.62 ± 2.32 | 87.53 ± 7.79 | 95.71 | 94.21 |
| Right Kidney | 91.64 ± 7.42 | 86.56 ± 10.75 | 89.93 | 89.99 |
| Spleen | 90.01 ± 11.75 | 86.46 ± 11.41 | 89.07 | 87.38 |
| Pancreas | 80.75 ± 6.11 | 75.89 ± 11.14 | 78.74 | 90.38 |
| Aorta | 91.19 ± 6.41 | 86.65 ± 10.95 | 93.10 | 95.77 |
| Inferior vena cava | 85.28 ± 6.67 | 74.10 ± 9.62 | 87.86 | 88.84 |
| Right adrenal gland | 77.26 ± 6.15 | 87.09 ± 6.49 | 75.28 | 88.97 |
| Left adrenal gland | 72.97 ± 12.22 | 89.94 ± 10.07 | 71.41 | 84.42 |
| Gallbladder | 77.52 ± 19.74 | 74.22 ± 21.76 | 73.76 | 71.39 |
| Esophagus | 71.32 ± 17.38 | 71.31 ± 15.57 | 74.86 | 86.97 |
| Stomach | 86.01 ± 10.75 | 76.32 ± 17.93 | 85.69 | 87.01 |
| Duodenum | 68.93 ± 11.93 | 61.00 ± 13.61 | 69.55 | 88.03 |
| Left kidney | 84.45 ± 20.49 | 79.58 ± 18.90 | 85.51 | 85.07 |
| Tumor | 12.49 ± 18.64 | 11.51 ± 15.44 | 11.83 | 6.69 |
| Average | 77.53 ± 11.28 | 74.87 ± 12.96 | 77.30 | 81.79 |

**Table 5.** Quantitative evaluation of segmentation efficiency in terms of the running them and GPU memory consumption.

| Case ID | Image Size | Running Time (s) | Max GPU (MB) | Total GPU (MB) |
|---|---|---|---|---|
| 0001 | (512, 512, 55) | 23.46 | 1694 | 17257 |
| 0051 | (512, 512, 100) | 19.13 | 1978 | 17698 |
| 0017 | (512, 512, 150) | 35.94 | 2562 | 28826 |
| 0019 | (512, 512, 215) | 23.33 | 1694 | 21224 |
| 0099 | (512, 512, 334) | 29.93 | 2564 | 26540 |
| 0063 | (512, 512, 448) | 37.86 | 1694 | 33508 |
| 0048 | (512, 512, 499) | 41.66 | 1978 | 37977 |
| 0029 | (512, 512, 554) | 52.81 | 1694 | 46037 |

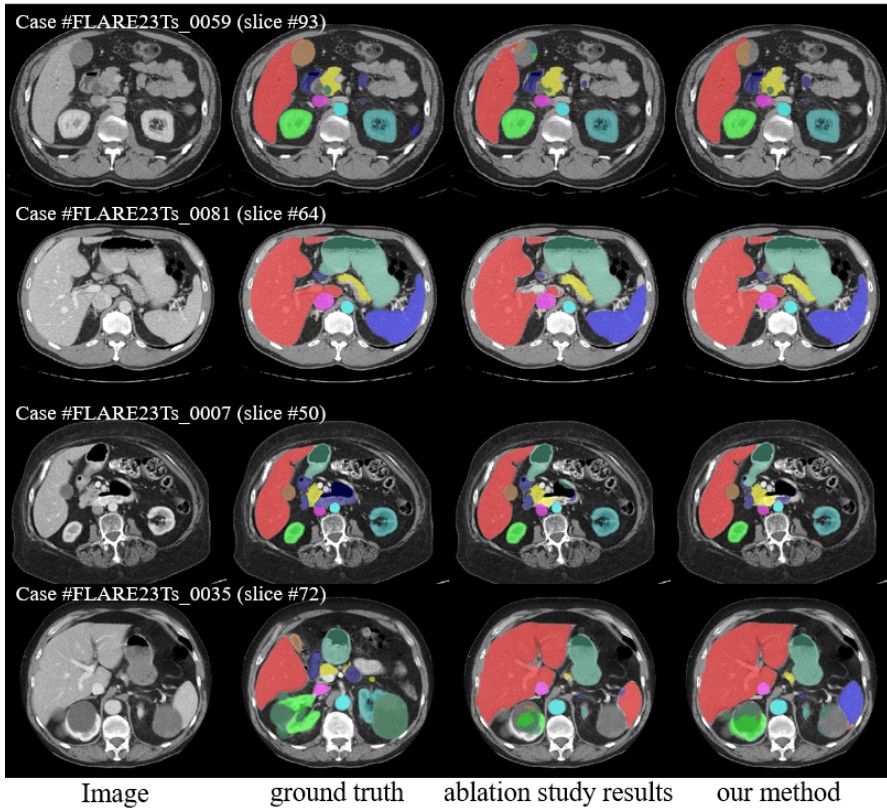

**Fig. 2.** The top two lines are good results, while the bottom two lines are bad results. Only labeled data are used in the ablation study.

### 4.3   Segmentation efficiency results on validation set

The segmentation efficiency results of eight cases in the validation set under the hardware environment provided by the organizer are shown in Table 5. Also, we calculated the average segmentation efficiency of all the cases, with the mean running time of 25.34 seconds, the max GPU memory of 2317MB and the total GPU memory of 23018MB. This is actually a good memory and time consumption, which can be attributed to the lower computational complexity of lightweight nnU-Net.

### 4.4   Results on final testing set

The results on the final testing set are given in Table 3.

### 4.5   Limitation and future work

As you can see, the evaluation metrics of our method are not high, especially in tumor segmentation scenarios. The reason for this may be that we have not fully utilized unlabeled data and have not utilized tumor information in unlabeled data. In the future, we will continue to work on this foundation and try to make more full use of unlabeled data.

## 5   Conclusion

In FLARE23 contest, we designed a model combining a lightweight nnU-Net and target adaptive loss, to segment all the organs and tumors in CT volumes and get a model trained based on the partially labeled dataset. Although the results we obtain are not that satisfying, this is the foundation of our future work and we will pay more attention to mking full use of unlabeled data and partially labeled dataset.

**Acknowledgements** The authors of this paper declare that the segmentation method they implemented for participation in the FLARE 2023 challenge has not used any pre-trained models nor additional datasets other than those provided by the organizers. The proposed solution is fully automatic without any manual intervention. We thank all the data owners for making the CT scans publicly available and CodaLab [17] for hosting the challenge platform.

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

**Table 6.** Checklist Table. Please fill out this checklist table in the answer column.

| Requirements | Answer |
| --- | --- |
| A meaningful title | Yes |
| The number of authors ($\leq 6$) | 4 |
| Author affiliations and ORCID | Yes |
| Corresponding author email is presented | Yes |
| Validation scores are presented in the abstract | Yes |
| Introduction includes at least three parts: background, related work, and motivation | Yes |
| A pipeline/network figure is provided | 1 |
| Pre-processing | 3 |
| Strategies to use the partial label | 4 |
| Strategies to use the unlabeled images. | 4 |
| Strategies to improve model inference | 4 |
| Post-processing | 4 |
| Dataset and evaluation metric section is presented | 5 |
| Environment setting table is provided | 1 |
| Training protocol table is provided | 2 |
| Ablation study | 6 |
| Efficiency evaluation results are provided | 5 |
| Visualized segmentation example is provided | 2 |
| Limitation and future work are presented | Yes |
| Reference format is consistent. | Yes |