# OpenReview forum: "A Lightweight nnU-Net Combined with Target Adaptive Loss for Organs and Tumors Segmentation"
_MICCAI.org/2023/FLARE — Submitted to FLARE 2023_

### Official Review · Reviewer_oXoc · 2023-09-19
**Reasonable Approach, Yet Refinement in Details Needed**

**Rating:** 6
**Confidence:** 5

**Review:**

# Summary

The paper introduces a model that combines a lightweight nnU-Net with Target Adaptive Loss (TAL) for efficient segmentation of abdominal organs and tumors.

# Strengths

This solution treats unannotated organs as background and employs the Target Adaptive Loss (TAL) for processing, effectively utilizing the partially annotated datasets. This strategy is both straightforward and practical.

# Weaknesses

- Figure 1 appears overly simplistic and does not adequately showcase the method's framework. It would be beneficial to enhance its aesthetics and incorporate more detailed information.

- In Figure 2, it would be helpful to clearly indicate what specifically the ablation study entails.

- In Section 3.1, there's an error in the citation related to nnU-Net.

- An ablation study comparing the use and non-use of the 1800 pseudo-labels is necessary. As you mentioned, since these 1800 images only have organ annotations, directly using these pseudo-labels might potentially degrade the performance.

---

### Official Review · Reviewer_v4zy · 2023-10-02
**The paper presents a comprehensive study on the application of nnU-Net in conjunction with a target adaptive loss function for abdominal organs and tumors segmentation tasks with a labeled and unlabeled dataset. The paper has meticulously detailed their methodology, including pre-processing, strategies for using partial labels and unlabeled images, model inference improvement.**

**Rating:** 6
**Confidence:** 5

**Review:**

Strengths:
1. The paper is well-structured and provides a clear and concise explanation of the methodology used.
2. The authors have done a commendable job in detailing the strategies for using partial labels and unlabeled images, which is a challenging aspect in image segmentation tasks.
3. The inclusion of an ablation study, efficiency evaluation results, and visualized segmentation examples adds credibility to the presented work.

Weaknesses:
1. The paper does not introduce any novel or groundbreaking techniques or methodologies for the task of abdominal organ segmentation. It relies on existing methods and combines them in a specific manner.
2. Although the proposed method provides a viable solution for handling partially labeled datasets, the achieved results do not stand out significantly when compared to other state-of-the-art approaches.

---

### Official Review · Reviewer_LcCR · 2023-10-25

**Rating:** 7
**Confidence:** 5

**Review:**

Sec 2.2: Please introduce your strategies to deal with the partial labels.

---

### Official Review · Reviewer_8UYG · 2023-10-25

**Rating:** 3
**Confidence:** 5

**Review:**

1. The figure resolution is low.

2. The description of each part of the article is too simple. Please provide more details.

---

### Decision · Program_Chairs · 2023-10-24

Accept

---

> ### Author Response · Authors · 2023-11-14
>
> Thank you very much for your careful review kind reminder. We have submitted our responses to the reviewers and our revised manuscript.